# Issues Regarding the Welfare of Assistance Dogs

**DOI:** 10.3390/ani12233250

**Published:** 2022-11-23

**Authors:** Amanda Salmon, Carlie Driscoll, Mandy B. A. Paterson, Paul Harpur, Nancy A. Pachana

**Affiliations:** 1School of Psychology, University of Queensland, Brisbane, QLD 4072, Australia; 2School of Health and Rehabilitation Sciences, University of Queensland, Brisbane, QLD 4072, Australia; 3School of Veterinary Science, University of Queensland, Brisbane, QLD 4072, Australia; 4Royal Society for the Prevention of Cruelty to Animals Queensland, Brisbane, QLD 4076, Australia; 5TC Beirne School of Law, University of Queensland, Brisbane, QLD 4072, Australia

**Keywords:** assistance dogs, welfare, regulations, support

## Abstract

**Simple Summary:**

This article considers the welfare of assistance animals and how this welfare can be maintained and monitored through a number of lenses. The article recognizes that the number of assistance animals in society is increasing as are their various assistance roles. What is meant by welfare and that it goes beyond the mere providing for basic needs is discussed using the Five Domains model of animal welfare. The legal instruments that in some way influence the rights of people with assistance animals including how these instruments do not, in the main, protect people against the barriers they may face, for example, when moving into aged care facilities are considered. The inability of people to keep their assistance animals when moving represents a reduction in the wellbeing of the person and also a poor outcome for the animal. The animal may end up in a shelter looking for a new home. Finally, the need to have processes in place to monitor the welfare of the animal, particularly when assisting people with a mental disability who may struggle to ensure that welfare, and support available for any person with an assistance animal who is struggling financially, is explored.

**Abstract:**

While the roles and efficacy of assistance animals have received attention in the literature, there has been less research focused on animal welfare issues regarding assistance animals. This is a pertinent area, given the burgeoning of types of assistance animals, situations in which they are employed, and access issues arising from increased assistance animal engagement. Animal welfare as pertains to assistance animals is discussed in this paper with respect to overall research on animal welfare concerns in a variety of contexts, training and access issues, and legal and regulatory concerns. Relevant examples from global contexts, as well as the specific Australian context, are offered. Conclusions include that while human quality of life is often considered and protected in laws and policies, this is much less true for assistance animals. Additional attention is required to ensure that the quality of life of both persons with disabilities and their assistance animals can be ensured. Support for a person to meet the needs of an assistance animal, as well as considerations for keeping such working dyads together in changed circumstances (e.g., following a move to an assisted living mode of accommodation), are recommended.

## 1. Introduction

Throughout the developed world, the appreciation of dogs as highly sentient beings, who possess value beyond the concept of possessions or workers, is being reflected through rapidly expanding representation in research endeavours, community actions, and legislative changes [1]. Concomitantly, awareness and commitment to the welfare of animals engaged in animal-assisted services, such as assistance dogs, has bloomed with the public now holding higher expectations of owners’ duty of care and lower tolerance for inadequate care conditions [2,3]. In fact, the paradigm of animal welfare itself is undergoing somewhat of a revolution, as the well-known Five Freedoms morphs into newer models with higher ideals defining the notion of welfare. This article discusses the welfare of assistance animals, particularly dogs, through a variety of perspectives, including increased research in this area; legislative, training and access concerns; and issues of animal welfare in a range of situations, including some illustrations from an Australian context.

## 2. Research on Assistance Animal Welfare

Assistance animals help persons with disabilities mitigate the impact of their impairments [4] and are employed in a range of public and private spaces [5,6,7]. A recent study by Gibson and Oliva [3] noted that while the Australian public was highly supportive of assistance dogs, there appeared to have been an increase since 2002 in the proportion of those expressing welfare concerns for these animals (7%), particularly in relation to the stressful and restrictive nature of the working role, the toughness of training, the lack of self-agency for the dog, and questionable ability of owners to provide for the dog’s needs. Now, more so than ever before, a social licence dependent upon animal welfare indicators is necessary for the sustainability of any animal-dependent human service [3], including assistance animals. 

Benefits accruing to the human owner/handler of assistance dogs are numerous and generally well-documented. In the case of Hearing Dogs, professionally trained canines who assist persons with hearing impairment by alerting them to environmental sounds for the purpose of independent living [8,9], additional gains from ownership include but are not limited to: the provision of companionship, a sense of security, stress reduction, decreased anxiety, reduced social isolation, increased physical activity level, and improved health-related Quality of Life [10,11,12,13,14]. However, a Hearing Dog comes with certain responsibilities that may directly affect the animal’s welfare. 

It is unknown at present, whether the care provided to assistance dogs matches or exceeds that extended to companion dogs, or even if such care for either type of animal by the public is typically sufficient for leading an optimal and flourishing life. Many aspects of assistance dog ownership may pre-dispose the animal to poor welfare states, including: lack of animal care knowledge in the owners; the high and often unsubsidized cost of purchasing, training, and caring for a dog (such as food, preventative medicines, routine and emergency veterinary care, insurance, etcetera [15]); the increased physical fitness and mobility requirements inherent in dog maintenance (the need for regular feeding, walking, bathing, and other care routines); inappropriate handling and task expectations; the constancy of work and minimization of downtime; inappropriate matching of animal temperament, capabilities, and needs with that of the owner and family; unregulated training and placement standards, as well as the perceived distinction between “working” dogs and “pet” dogs.

Compounding the risk of negative welfare states for assistance dogs is the scarcity of research literature addressing the unique welfare needs of dogs with occupations beyond military and farming contexts [4]. This remains particularly true for Hearing Dogs, Mobility Dogs, and animals with a remit of providing psychiatric or medical-alert assistance. Furthermore, whilst the guidelines and position statements of organizations associated with assistance animals inevitably include mention of the animal’s welfare being a primary consideration (e.g., [16]), very few make mention of the animal’s right to flourish or to lead a life worth living where mutual benefit exists for both animal and human. Additionally, the ethical principles of Nonmaleficence and Beneficence (see [17]) are most certainly not present in animal-related legislation, where at a local level, the Queensland Government Guide, Hearing and Assistance Dog Act of 2013 [18] fails to include any sections pertaining to dog welfare.

Awareness of the basic needs of companion and assistance dogs is likely of a reasonable level within many developed countries, yet these traditional conceptions have historically focused on welfare as defined by physiological ill health and stressors, in line with the well-known Five Freedoms [19], rather than the animal’s psychological and overall wellbeing. Mellor et al.’s [20] Five Domains model offers a method of understanding both the physical and mental effects of working on Hearing and other assistance dogs. This model acknowledges that the human–animal relationship is an important determinant of animal welfare, in line with accumulating evidence that demonstrates that positive relationships are rewarding to the animal and lead to improved welfare [21], as well as long-term stress resilience and protection against disease. The model’s overriding welfare objective is to promote positive experiences between human and animal, instead of merely minimizing potential for negative states. Hence, most importantly, the model promotes conditions in which the animal is encouraged to prosper, in spite of and due to their working role. 

Future research concerning the welfare of assistance dogs may utilize the modern Mellor model in a manner similar to Fletcher et al.’s [20] investigation of UK horse owners’ perceptions of wellbeing. Such would allow eventual development of practical welfare assessments, validated through congruence with physiological and behavioural indicators, that would promote best practices in assistance dog welfare.

## 3. Regulatory Issues

A considerable amount of research and regulatory attention has focused on the benefits of assistance animal ownership for humans [22]. There is evidence that the protections afforded to animals that support persons with disabilities have insufficient research and regulatory attention, and that the shifts in assistance animal engagement has heightened the need to ensure the animals that serve their handlers are protected. 

For a considerable period of time, the disability assistance animal landscape was stable. Disability assistance animals were provided by charities supporting the blind, deaf or mobility impaired, and those charities in turn ensured that dogs were trained in a humane way and monitored the welfare of the dog throughout its working life. Illustratively, for decades, almost all reputable charities that provided disability assistance animals across the globe were members of either the Assistance Dogs International or the International Guide Dog Federation. Both associations ensured that their members protected animals as a requirement to remaining members. Illustratively, Standard 4 of the International Guide Dog Federation membership addresses the “humane care” of animals [23], and Assistance Dogs International ensures the physical and emotional safety of assistance dogs by “training methods, care and treatment that demonstrate partnership, appreciation and respect” to the dogs that they work with [24]. These charities have always relied heavily upon public donations. Accordingly, these organizations have always had a strong incentive to ensure the welfare of animals. There is a growth in disability assistance animals being sourced outside such established charities, with groups and individuals sometimes acting fraudulently in this space [25].

Relatedly, persons who are blind, hearing- or mobility-impaired rely upon their assistance dogs for their daily activities. Animals who are not well treated become unable to provide these services. the capacity to see, hear or walk does not impact upon the care for animals. Accordingly, handlers themselves have the motivation and ability to protect their animals. 

Consequently, most regulatory interventions that focus on protecting disability assistance animals, aim to protect the animal from bad actors, such as criminals who attack persons with disabilities that “assault(s) any person who relies on a guide, hearing or assistance dog(s)” [26], or perpetrators of domestic violence [27,28,29] or irresponsible dog owners who do not keep their pet dogs on leads in public and cause harm to others through their neglect [30]. 

As well, there has been growth in the breadth of disabilities that utilize assistance animals [31]. Whereas dogs who replace a sense or physical capacity that has no impact on the capacity to care for oneself or an animal, animals that help replace a cognitive process can be placed with handlers who have their capacity to care for themselves and animals reduced [32]. For example, there has been growth in the area of provision of assistance dogs for persons living with dementia, but the research in support of specific benefits is mixed [33]. Marks and McVilly, in their 2020 review paper [33], specifically note the lack of consistency in the quality and training of dogs for persons living with dementia. In some instances, it was unclear whether the dogs had even received any specific training. Given that one goal of the provision of an assistance animal is to improve the wellbeing of the recipient, it would appear that poorly trained animals would carry a high risk of actually decreasing the wellbeing of both the person living with dementia and their care partner(s).

Barstad [34] reports on an evaluation of the welfare of dogs working with animal assisted interventions for older persons living with dementia. The empirical study involved examining stress-associated behaviours in 13 dogs participating in animal assisted therapy or animal assisted activities. While the dog’s comfort and welfare was found to be relatively high during such activities, a comparable study on service dogs’ welfare while living and working in the care of persons living with dementia and their care partners is currently lacking. 

Similarly, there is an increase in assistance animals for persons with a psychiatric disorder [35]; again, the literature in this area is relatively small. As with persons living with dementia, many psychiatric assistance dogs are trained either by the owner or a dog trainer, rather than being provided under the auspices of a charity or mental health organization [35]. The largest evidence base within this category of assistance dogs has been those trained to work with persons with post-traumatic stress disorder (PTSD) [36]. This literature is also increasingly including guidelines for the training and care of dogs for this population [37]. Specific research on welfare concerns for animals working with those with PTSD is lacking but has been highlighted as a concern and area for future research [37].

Such new applications for disability assistance animals requires increased regulatory attention to animal welfare. We use the term regulation here to include any legal or policy interventions which can achieve the desired result [38]. 

Regulatory interventions which target disability assistance animals must comply with the equality paradigm in the UN Convention on the Rights of Persons with Disabilities (CRPD). The CRPD is built upon the principle of equality [22]. Importantly, this means that laws and policies should not impose an additional burden upon people because they have a disability. Thus, requiring persons with disabilities to prove they are not endangering animals, without a very good cause, is contrary to the equality principle in the CRPD. After all, it is reasonable to presume that a person who is blind and uses their guide dog as their eyesight is probably more likely to care for that dog than a person with no disability who has a pet in their backyard. 

The way around this is not to impose a condition because a person has a disability, but instead build on existing approaches around where a handler/animal relationship gains increased statutory protections. 

The Federal Disability Discrimination Act 1992 (Cth) s 9 defines an assistance animal to include an animal that is trained to assist a person with a disability to alleviate the effect of the disability, which also meets standards of hygiene and behaviour that are appropriate for an animal in a public place [39]. Section 8 then explains that discriminating against an assistance animal is discriminating against a person due to their disability, and Sections 5 and 6 prohibit direct and indirect discrimination.

Section 54A provides protection for those who desire to ensure that an animal is, in fact, a disability assistance animal by rendering it lawful for a person controlling a public space to ask for evidence. The evidence requested extends to asking about the hygiene and infections of the animal, but does not concern animal welfare beyond this. Australian Federal laws unfortunately provide no guidance on what evidence is reasonable, nor does it provide a framework for demonstrating that the handler/animal team gain protection.

Proving the status of an assistance animal is made easier in some State jurisdictions by the operation of a government issued identity card showing that the handler and animal are, in fact, recognised as a protected handler team. Confusingly for many, including for those attempting to implement the law, a person can assert their federal rights without gaining certification from a State jurisdiction that their animal is protected or cared for appropriately. Handlers may simply decide not to have their animal certified, or the person may fall through a gap between coverage, where the federal law applies to all animals, whereas the state laws, such as those in Queensland, only apply to dogs [22]. Additionally, for dogs such as those being provided to persons living with dementia, there is no such recognized certification available.

The Australian State of Queensland has one of the more developed assistance animal identification processes. To receive a handler card in Queensland, an animal-handler team must become certified by an approved trainer as being either a guide dog for the blind or deaf or an assistance dog under Part 4 Division 2 of the Guide, Hearing and Assistance Dogs Act 2009 (Qld) [40]. A requirement for certification is that the dog can perform the support tasks on an on-going basis. It is implicit in this requirement that the dog is treated in a way that enables it to continue to perform the service for which it is certified. This position is supported by the fact that Section 46(2) enables the handler/dog certification to be cancelled if the dog for whom the card was issued should be retired because of age, illness or other inability to be employed as a guide, hearing or assistance dog. This cancellation process is usually initiated by a trainer working with an approved certification body. If a handler loses their certification, this however does not prevent them from asserting under federal laws that their dog is protected as a disability assistance animal.

## 4. Access Issues for Service Dog Users in Later Life

The federal Disability Discrimination Act 1992 [41] and state acts, such as the Guide, Hearing and Assistance Dogs Act 2009 [18] ensure that certified assistance dogs and their owners maintain access rights. This includes accommodation and incorporates both refusal and stipulations that would require owner-dog separation. Despite this, many aged care facilities continue to disallow owners to retain their assistance animal, and proving disability to obtain an assistance animal while in aged care has considerable barriers [42]. A report by the Animal Welfare League Australia (AWLA) found that of 2933 aged care facilities reviewed, only 18% considered allowing residents to keep an animal [43]. While this did not pertain specifically to assistance animals, it is indicative of the difficulties with relocating to an aged care facility with an animal. 

Assistance animal ownership has been found to be highly beneficial for older adults, not just from a practical perspective, but across social, mental, and physical health domains [44]. As such, it is ideal that older adults do not avoid obtaining an assistance animal due to concerns over future retention in the case of relocation into aged care. Further, for those who already own an assistance animal, that they do not need to experience feelings of loss associated with not only the separation itself, but the loss of the relationship and benefits the animal provided [45]. Particularly as they are already likely experiencing compounding losses downsizing, losing their home, and in some cases their autonomy, this situation can be quite devastating.

Consideration of the animal itself is also often overlooked when they are required to be relinquished. In the case of assistance dogs, owners are generally able to contact their provider to return the dog [46]. However, where the animal is self-trained by the owner, in the absence of a suitable friend or family member, they may need to be relinquished to a shelter, such as the RSPCA. Additionally, the process of relinquishment is not only upsetting to the owner but may be considerably distressing and confusing to the animal itself. 

While there is little research on the reasons aged care facilities may prohibit assistance animals, it can be extrapolated that they are the same reasons as for companion animals. These include: the age of the owner, lack of support funding and staff to assist in animal maintenance, and health and safety concerns [43,47]. It is often assumed that the advanced age of the owner can affect their ability to provide sufficient care to their animal, or that their animal will out-live them. However, it has been argued that lack of sufficient animal care or the death of an owner could happen at any age, for a range of reasons, such as unexpected death or illness, or competing life interests [47]. Thus, suitability of continued ownership should be based on the health and abilities of the owner, rather than age, which is discriminatory. 

On consideration of the aged care facility’s lack of support funding and staff who are able to assist, where owners are able to care for the animals themselves there should be limited assistance required. Further, assistance animals provide practical supportive utility, such as mobility and sound alerting, and animals broadly have been found to improve mental and physical health aspects when introduced into aged care facilities [48]. As such, it is possible that the owner would require less support from staff from a practical perspective, as well as less emotional and physical support requirements. This may also impact the amount of funding required due to maintained health. 

Health and safety concerns in any aged care setting are of the upmost importance, both for staff and residents. While some free-roaming companion animals may cause issues such as a trip hazard, assistance animals are specifically trained and, thus, less likely to create these issues. Zoonotic illnesses, parasites, and contamination related issues are possible. However, these can be limited through regular animal vaccination, hygiene practices, and cleaning [49]. It is also important to note that the potential benefits for the owner outweigh these limited risks. 

Overall, the current antidiscrimination legislation is designed to protect assistance dog owners and allow them the same access to sufficient care and accommodation. While aged care facilities often do not allow animals of any kind, a reconsideration of these rules for assistance animals is crucial, particularly, as it can have the ramification of older adults deciding not to obtain an assistance animal if they expect to need to relocate to an aged care facility in the future, despite the significant benefits of ownership.

## 5. Assistance Animal Welfare through the Eyes of an Animal Welfare Organization

The Royal Society for the Prevention of Cruelty to Animals (RSPCA) runs shelters throughout Australia and receives animals from a number of sources including strays, owner surrendered animals and animals seized due to cruelty or neglect. The proportion of each of these categories within shelter intakes varies with shelter and location, and over time. For example, in 2014 in Queensland, only 19% (2279) of incoming dogs were owner surrender while 58% (6997) were stray and 6% (749) seized [50]. For cats, the proportions were similar [51]. Over the last few years, intake numbers have decreased significantly in both the US [52] and Australia, and the proportion of intake animals surrendered by owners has increased [53]. 

Of interest for assistance animals generally are the numbers surrendered by owners or who enter the shelter through the animal inspectorate arm of the RSPCA. Firstly, considering relinquishment, several studies report that the majority of dogs are surrendered for owner related reasons [54,55,56,57]. A review by Coe et al. [58] reported that 81% of the research papers located (*n* = 84) reported owner-related reasons were the major causes of dog relinquishment. 

Housing issues, including moving and not being able to have the dog in new rental accommodation, are well recognized [54,56,57,59,60]. Owner health reasons, including both mental and physical health, are also commonly reported [54,57]. Finally, financial problems were identified in several studies, particularly those that involved interviews with relinquishes [55,60,61]. No study was located that specifically looked at the relinquishing of assistance dogs, but considering the listed reasons, it is impossible to imagine that people with an assistance dog do not face these same issues—moving and not being able to take the animal, deterioration in health including mental health such that it becomes difficult to care for the animal and, finally, not having the finances necessary to continue to care for an animal. 

With respect to inspectorate seizures, a retrospective study of calls received by the RSPCA Qld from 2008 to 2018 revealed that the majority of the complaints received were neglect-related rather than deliberate cruelty [62]. The most common complaints related to poor body condition resulting from insufficient food and water, poor living conditions, and insufficient care and exercise. Additionally, the dog owners tended to be of lower socioeconomic status than the median for Queensland [63]. In many cases the neglect is due to ignorance, forgetfulness, or temporary lapses, and should be viewed as a marker for co-occurring self-neglect [64] often indicating deteriorating mental health. Given the above discussion, it is likely that assistance animals also enter shelters for these reasons. Such cases can often be handled through physical and psychological support by, for example, providing food and veterinary care to the animal, and psychological support for the owner. 

RSPCA Qld has no record of the number of assistance or support dogs that are either surrendered or admitted through the inspectorate. However, given the reasons discussed above, it is probable that some of the dogs, particularly older animals, that end up in our shelters were assistance or support animals that can no longer stay with the owner. 

The importance of keeping these animals with their owners in most cases must be stressed. Pinillos [65] argues that the use of the One Welfare framework can help to prevent assistance animals from being seen as a burden, and moves the focus to providing support, to keep the owner and animal together. This support maintains the benefit for the human needing assistance and represents better welfare for the animal. This, in turn, benefits the wider community. Therefore, supporting people needing assistance and the assistance animals directly should be the aim of everyone involved.

## 6. Conclusions

Assistance animals increase the quality of life enjoyed by people with various disability types. As a result, laws and policies seek to protect the right of people with a disability to be accompanied by an assistance animal. In contrast to protecting a person’s quality of life, laws and policies fail to address the quality of life of assistance animals. As a result, we argue additional attention is required to ensure that the quality of life of both persons with disabilities and their assistance animals can be ensured. Support may be required to keep the animal and person together including changes in accommodation laws to make it easier for people to find suitable and affordable accommodation where they are allowed to have pets. That is, the present barriers including the application process to be allowed to have pets in accommodation must be removed or reduced. Support for the person to meet the animal’s needs when required should also be available (basic food needs and veterinary care), so they are not required to relinquish their animal for financial reasons.

## Data Availability

Not applicable.

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
