# Peer review of "Issues Regarding the Welfare of Assistance Dogs"

_animals, 2022, doi:10.3390/ani12233250_

Round 1
Reviewer 1 Report
This article is interesting and very necessary, as there is little research on the welfare of companion animals.
Author Response
Please refer to the attached. We thank the reviewer for their feedback and taking the time to review our article.

Reviewer 2 Report
Overall: Given the pervasiveness of incorporating animals in a variety of conditions to help with humans, there is a larger need for discourse relate to how the animals adjust to these circumstances and how their welfare and wellbeing needs are being met. As the authors point out, there is adequate research in many fields regarding the benefits to humans, but there is a lack of research into the wellbeing of animals in these roles. Exceptional paper.
Simple Summary and Abstract: Very clear and concise explanation of the purpose of the article and relevance to not only the wellbeing of the animal, but also the consequences to the field and working dyad.
Introduction: Short, but appropriate for this type of article. It sets the reader up with the circumstances of the current climate of welfare in assistance animals.
Research on Assistance Animal Welfare: This is a good overview of the existing focus of research on the human benefits with much less on the welfare and wellbeing of animals in these roles. The authors do a good job of contrasting the research done on human outcomes compared to the lack of research in wellbeing of the animals involved. The authors also do a good job of briefly, but adequately explaining how expectations on these animals can lead to both physical and psychological concerns.
Regulatory Issues: Again, the authors do a good job of explaining the general context of the regulations, especially related to animals in services related to hearing and seeing and the history of the field. The authors also do a good job of describing how the welfare of the animal and human are integrated, further supporting the need to realize welfare and wellbeing concerns at a deeper level.
The authors also do a good job of constructing arguments for the support of wellbeing of animals while still supporting equality and CRPD requirements.
Access Issues for Service Dog Users in Later Life: The authors do a great job of explaining the complexities around the integration of service dogs into different environments for those needing them at later stages in life, including in residential facilities. It’s especially nice to see that they suggest that animals in these environments might decrease the load on staff and therefore potentially improve wellbeing for all involved.
Assistance Animal Welfare Thought the Eyes of an Animal Welfare Organization:
Line 277 – Perhaps the word “thought” was mean to be “through”?
Overall, this perspective is also important with regards to the topic of welfare and wellbeing in service animals. Recognizing the role of relinquished animals is critical in emphasizing the need to look more closely at how animals can remain in their homes and with their teams.
Author Response

(The authors gave the same response as above.)

Reviewer 3 Report
Overall – a very important topic and one that has not received sufficient attention. I know from personal experience that nursing homes vary greatly in their rules with respect to animals and your conclusions lines 246-276 are quite correct.
The manuscript is written in a good style, which is easily accessible to readers.
Comments:
· Lines 72-78 contain a fair amount of material describing issues – it would be helpful if the sources for these statements were referenced.
· Lines 85-86 – I agree with the conclusion about flourishing etc – but is this part of the welfare paradigm? The authors refer to the Five Domains model around line 95 - maybe introduce something on this earlier (in the introduction?) – just a sentence or two, so the reader has a feel for the way the manuscript is structured
· Lines 87-89 Qld regulation being introduced as an example – perhaps make this clearer. Reason this is suggested is that manuscript draws examples from a few jurisdictions (eg horses in UK – line 106, Assistance Dogs International line 125) yet introduction in line 46 says that illustrations come from Australian context.
· I really like the discussion in lines 156-161 – and this is important given that dementia cases are set to rise with an aging population, which makes the lack of research identified by the authors even more important.
· Lines 192-195 – would state/territory anti-cruelty laws etc operate in this field and if so, are they adequate. I suspect they are not which is why the authors are arguing for specific regulation. OK – deal with it in next parags. I have attached some cases on service animals – they may be useful and if not, please ignore.
· Section 5 that deals with assistance animal welfare through the ese of animal welfare organisation
o Typo in line 277 “thought” instead of “through”
o Suggest that material around line 314 be brought to beginning of this section. As I was reading I thought that material lines 278-313 was good, but did not specifically relate to assistance animals. However the statement that “it is probable that some dog, especially older ones were assistance animals, needs clarification (lines 315-6) – is this a conclusion drawn from shelter staff or something else?
Grammatical/spelling issues
· Line 54, split infinitive “to adequately provide”
· Line 55 “license” – if American spelling is used in the journal, that is fine

Author Response

(The authors gave the same response as above.)
